# Guided Endodontic Surgery: A Narrative Review

**DOI:** 10.3390/medicina59040678

**Published:** 2023-03-29

**Authors:** Azhar Iqbal, Thani Al Sharari, Osama Khattak, Farooq Ahmad Chaudhry, Alzarea K. Bader, Muhammad Mudassar Saleem, Rakhi Issrani, Ibrahem T. Almaktoom, Raghad Fayez H. Albalawi, Ebtehal Dhyab M. Alserhani

**Affiliations:** 1Department of Restorative Dentistry, Jouf University, Sakaka 72388, Saudi Arabia; 2Department of Restorative and Dental Science, Faculty of Dentistry, Taif University, Taif 11099, Saudi Arabia; 3Edinburgh Medical School, Department of Medical Education, The University of Edinburgh, GU, 316 Chancellor’s Building, Edinburgh EH16 4SB, UK; 4School of Dentistry, Shaheed Zulfiqar Ali Bhutto Medical University (SZABMU), Islamabad 44000, Pakistan; 5Department of Prosthetic Sciences, College of Dentistry, Jouf University, Sakaka 72388, Saudi Arabia; bkzarea@ju.edu.sa; 6Consultant Oral and Maxillofacial Surgeon, Ministry of Health, Dental Center, Yanbu General Hospital, Yanbu 46411, Saudi Arabia; 7Department of Preventive Dentistry, College of Dentistry, Jouf University, Sakaka 72388, Saudi Arabia; 8College of Dentistry, Jouf University, Sakaka 72388, Saudi Arabia

**Keywords:** endodontic microsurgery, guided surgical endodontics, cone-beam computed tomography (CBCT), endodontic therapy, 3D guide

## Abstract

*Background and objectives*: Endodontic surgery has evolved over the last two decades. The use of state-of-the-art guided endodontic surgical procedures produces a predictable outcome in the healing of lesions of endodontic origin. The main objective of this review paper is to define and characterize guided surgical endodontics as well as its benefits and drawbacks by reviewing the most recent relevant scientific literature. *Methods*: A literature search was conducted using multiple databases comprising of MEDLINE (via PubMed), EMBASE, and Web of Science. The terms used for the search were ‘guided endodontics’, ‘surgical endodontics’, and ‘endodontic microsurgery’. *Results*: In total, 1152 articles were obtained from the analysis of the databases. Unrelated articles from the available full text of 388 articles were excluded. A total of 45 studies were finally included in the review. *Conclusions*: Surgical-guided endodontics is a relatively new area of study that is still maturing. It has many applications such as root canal access and localization, microsurgical endodontics, endodontic retreatment, and glass fiber post removal. Additionally, it does not matter how experienced the operator is; the procedure can be completed for the patient in less time and provides greater accuracy and safety than conventional endodontics.

## 1. Introduction

Traumatic injuries can result in increased dentin formation which leads to pulp obliteration (PO). This is due to calcification of the canal; however, it is not typically a sign of pulp damage. The other possible pulp response to injury is calcific metamorphosis, which manifests as a fast accumulation of firm tissue in the pulp region [1,2]. If a substantial amount of mineralized tissue has been deposited, the pulp space may completely disappear on radiographs, yet the histological sections may show some pulp tissue. Pulp obliteration can range from complete, where the pulp chamber and root ducts are indistinguishable, to partial, where the pulp chamber is still visible but the root canals are noticeably thin but still visible [3]. Inflammation of the pulp is a healing response that reveals the tooth’s ability to repair, whereas the death of the pulp is pulp necrosis (PN). This may result in damage to the supporting periodontal tissues which may become infected via the dentinal tubules [4]. There is evidence that PN is common among teeth that have been damaged by trauma [3,5]. The best treatment option for these teeth is unclear. Several publications have argued that PO is due to the increased likelihood of PN following trauma or therapeutic procedures like orthodontics and restorative procedures [6]. Nonetheless, this standard is already being disregarded. Endodontic treatment should be delayed until either clinical symptoms appear or periapical tissues are involved [7]. However, these teeth require close clinical and radiographic observation. The endodontic treatment of PO teeth can raise concerns at any time [8]. When employing aids such as magnifying glasses, microscopes, or cone beam computed tomography (CBCT) to guide treatment, the operator, especially a novice, may find it difficult to grasp the CBCT visuals, build a mental guide, and simultaneously conduct the therapy. In response to these challenges, the discipline of guided endodontics (GE) emerged. Guided endodontics is founded on digitally planning endodontic therapy. Root canal perforations and other iatrogenic complications can be avoided with the use of GE procedures [9,10].

In 2016, an innovative method utilizing 3D-printed guides or splints, previously utilized in implant procedures, was introduced to endodontics. The effectiveness of these techniques for root canal access was originally studied by researchers and positive findings were reported [11,12]. The next year, the first case report describing the successful use of this technique to treat teeth with PO was published [13]. The integration of GE (a printed surgical guide) with CBCT images represents a novel method for achieving root tip access in surgical endodontics. In this way, the surgeon is able to make more exact incisions in the gingiva and bone, remove roots with greater accuracy, and facilitate more rapid healing following the operation. Less time is needed for this type of treatment compared to improvisational methods [14,15].

These days, endodontic procedures can be carried out using either static (SGE) or dynamic guidance (DGE). The upper or lower arch is scanned using a CBCT during SGE. When the two pictures are superimposed in software, a template can be made to go over the desired tooth (and some adjacent teeth). This will be used as a template for a drill hole of the right size and orientation to break through the calcified canal’s walls. The root canal can then be accessed more precisely and steadily with a drill by fabricating cylindrical or sleeve-like guides. The inner, smaller cylinder is made of metal and fits snugly into the larger, outer cylinder. A stereolithography file is exported from the planning tool and used in a 3D printer to create the blueprints. A rubber dam is placed around the patient’s teeth, and the guide is tried in to make sure it fits properly before the procedure continues. Once the calcified tissue is removed using the metal cylinder as a guide, root canal therapy can proceed normally [16,17]. A stereo camera connected to an active navigation system can precisely time the entry and departure points when drilling into the pulp chamber or root canal. The operator can then watch their movements and fine-tune the position of the instruments [11,18,19,20].

The use of digital tools in addition to conventional endodontic planning is essential to the field of guided endodontics. In light of the fact that these methods have recently surfaced as a prospective treatment option for PO, a prevalent issue in endodontics, it is important to conduct a literature review to determine their advantages and disadvantages, as well as other possible applications. Against this background, this review paper was undertaken to define and characterize guided surgical endodontics as well as to highlight its benefits and drawbacks by reviewing the most recent relevant scientific literature.

## 2. Materials and Methods

This review is in accordance with the Preferred Reporting Items for Systematic Review and Meta-Analysis (PRISMA) 2020 Statement in order to maintain a codified organization of the study [21].

The search databases included MEDLINE (via PubMed), EMBASE, and Web of Science. The terms used for the search were ‘guided endodontics’, ‘surgical endodontics’, and ‘endodontic microsurgery’. The search terms were used to ensure that all the relevant papers were identified. Further searches were performed in the reference lists of relevant studies and in the literature reviews dealing with the topic of interest.

Considering the eligibility criteria, only the articles relevant to the topic of guided endodontic surgery were evaluated as suitable for inclusion in this review. Only papers published within the last 5 years (31 August 2017 to 1 September 2022) and written in English language were considered. Additionally, only original papers, case series, and case reports were considered. Interim reports, abstracts, letters, short communications, and chapters in textbooks were discarded. This eligibility criteria helped us identify papers related to the area of research.

All the studies resulting from the search strategies were imported into an Endnote library and duplicates were removed. Two reviewers (1st and 2nd authors) independently assessed the records (title and abstract), selecting the articles that met the eligibility criteria. Any type of disagreement was resolved by consulting a third independent reviewer (3rd author). After this screening, the records selected were analyzed in their full-text version, and two other reviewers (1st and 2nd authors) independently assessed whether they should be included in the review. In case of disagreement, a third author was consulted (3rd author). The same two reviewers carried out the extraction of the data in a standardized data form.

The PRISMA flow diagram (Figure 1) was used to report the included articles according to the eligibility criteria and those excluded during the study selection process.

## 3. Results

In total, 1152 articles were obtained from the analysis of the databases, adopting the search strategy described in the “Materials and Methods” section. Unrelated articles from the available full text of 388 articles were excluded. The articles which were not considered were those which had no relevance to guided surgical endodontics. The other reasons for excluding the articles were: they were short communications, comprehensive reviews, and policy statements. A total of 45 studies were finally included in the review which were related to our research topic.

**The studies’ characteristics:** Table 1 and Table 2 show the results of 45 studies that aimed at the efficacy of endodontic surgical treatments; 29 of these studies focused on SGE and 16 on DGE. There were a total of 45 studies; only 1 compared SGE to DGE, and the results were positive for both [22]. Twelve trials that compared GE (DGE or SGE) to conventional manual endodontic therapy found that GE was superior in terms of patient outcome [23,24,25]. Eight studies employed 3D reproductions to simulate genuine teeth while just one used plastic dentition [17,26,27].

We investigated how to guide apical access during endodontic microsurgery, how to remove fiberglass posts, where to find the more significant palatine artery to avoid damaging it, how to perform an osteotomy and apicoectomy, how to evaluate the technique in terms of fracture strength of the teeth, and how to evaluate the effectiveness of endodontic microsurgery. These aspects were investigated as they are related to guided surgical endodontics.

Commonly measured outcomes of SGE and DGE applications were how well they worked, how accurately they removed tissue, how much tissue was removed from teeth, and how quickly they treated patients.

At least 25 of the 45 reported cases were followed for at least a year, and all of them either experienced a complete resolution of their symptoms or evidence of bone regeneration [3,16,28,29,30,31]. Most research comparing SGE and DGE has concentrated on three aspects; efficacy (how well the procedure works), accuracy (how much tooth tissue is removed), and speed (how quickly treatment may be completed). A total of 15 out of the 45 case reports were followed for at least a year, and all showed either complete resolution of symptoms or indications of bone regeneration [32,33]. These procedures involved single-rooted teeth solely. Thirty-three of these cases involved pulp obliteration, eight involved osteotomies and apicoectomy, two involved re-treatments or one requiring the treatment of a thick evaginatus, and one involved the removal of fiberglass posts [9,12,34,35,36,37].

**Table 1 medicina-59-00678-t001:** Research articles included in the review information.

Previous Work	GE Research Topic and Classification	Origins and Varieties of Teeth	Operator Characteristics and Skill	Conclusions
Gambarini, G., et al., 2020 [38]	Incredibly efficient DC gear from DGE against MAN for AC precision	Type 2.6 R3D Ray Tracing Rendering	Operators fluent in both groups	Less tissue is lost during DGE, decreasing the possibility of iatrogenic coronary weakness.
Jain, S.D., et al., 2020 [34]	Root canal attrition in PO teeth (DGE vs. MAN)	Modeled R3D with PO 2.1 and PO 4.1	One EC (with a microscope for MAN access)	Compared to PO, DGE requires less tissue removal and provides more precise duct location.
Loureiro, M.A.Z., et al., 2020 [39]	California’s tooth extraction rate (SGE vs. MAN)	Molars in the mandibular midline and on the jaw’s sides are NAT I features (1st and 2nd jaws).	EC (with magnifying glasses)	SGE helps keep more molar tissue intact. When comparing the amount of tissue extracted from the incisors, there was no discernible difference.
Connert, T., et al., 2021 [13]	Root canal attrited teeth wearing down (miniaturized DGE vs. MAN)	Maxillae (both central and lateral) and cheekbones (R3D) in position C	One operator with 12 years of experience	Smaller DGE allows for more precise CA with minimal tissue sacrifice. There was no discernible difference in performance between inexperienced and experienced operators.
Koch, G.K., et al., 2022 [40]	Compare 3D printers (for SGE)	R3D All types	One EC	Difference between printers; Produced accurate guides for the ACs.
Buchgreitz, J. et al., 2019 [41]	DGE’s accuracy (teeth with apical periodontitis, teeth requiring post and with PO)	I (Central and Lateral) NAT	--	DGE is accurate in identifying canals with PO
Torres, A., et al., 2021 [32]	Pinpoint accuracy and reliable outcomes (with DGE)	R3D I (Central), PM mandibular and maxillary	One ES One EC	Incorporating a laser into DGE is a good idea for precisely slicing through sturdy dental tissues.
Strbac, G.D., et al., 2021 [42]	Preciseness in attrited canal, disturbances in linear and angular dimension during alternating current (with SGE)	NAT I, C, PM, and M.	--	SGE accuracy acceptable in AC M has angular and linear dispersion.
Meda, R.G., et al., 2022 [43]	Two-AC-Program Accuracy and Effort	It is a R3D model with a simulated PO.	One operator	The milling guide might be quickly planned using any program. Predictability in SGE treatment. PO teeth can safely undergo root canal localization.
Choi, Y., et al., 2021 [44]	Guidelines for preventing tooth loss effectively in California (student-oriented) (with SGE)	NAT PM and M	One ES pre-doctoral	In PM, students who employed the AOG-3DP guide saw a 74.9 percent reduction in preparation time, and in M, an 81.1 percent reduction. Methods that follow such advice more conservatively. Design and production take time. Even in more complex situations, the handbook can be used as a resource.
Simon, J.C., et al., 2021 [45]	Variation in fracture strength due to method (SGE vs. MAN) SGE’s effectiveness in treating CA with MTAs	NAT PM mandibular	One operator (with magnifiers for MAN)	SGE makes the process of removing MTAs quicker and less prone to mistakes. Increased resistance to tooth breakage is maintained with SGE.
Mediavilla Guzmán, A., et al., 2019 [46]	Efficiency and precision while using PO to locate ducts (DGE vs. MAN)	PO with maxillary and mandibular NAT I, C, and PM	One EC One ES	To find ducts with PO, DGE is more precise and effective.
Fan, Y., et al., 2019 [47]	Simulated PO for minimally invasive localization of attrited canal and channels (in DGE)	R3D All types	One EC	The average 2D horizontal deviation with DGE and high-velocity drills for canal localization with OP is 0.9 mm, whereas the average 3D deviation is 1.3 mm, and the average 3D angle deviation is 1.7 mm.
Smith, B.G., et al., 2019 [48]	Position of the palatine artery with reference to the molar apices on the efficiency of OT and AP. Flapless palatal procedure evaluation (with SGE).	Using CBCT scans of patients’ first and second maxillary molar teeth	Two EC	A total of 47% of upper first molars and 52% of upper second molars with a 2 mm safety margin can have endodontic surgery. Half of the maxillary first and second molars can be reached without a palate flap.
Aldahmash, S.A., et al., 2022 [29]	For OT and AP, the efficacy and precision of DGE (DGE vs. MAN)	For all NATs	One EC (with a microscope for MAN)	DGE improves accuracy over the MAN method. The effectiveness and precision of the MAN method are diminished when the roots are further away from the cortical bone.

(MAN: Manually, R3D: 3D replicas, I: Incisor, C: Canine, PM: Premolar, M: Molar, NAT: Natural, ES: Endodontic student, EC: Endodontic consultant).

**Table 2 medicina-59-00678-t002:** Other articles in the review.

Authors	Tooth	Previously Used Therapy	Trauma	Problem	EG Subtype	Results
Todd, R., et al., 2021 [6]	2.1	No	No	PO	SGE	No symptoms after 24 h
Buchgreitz, J. 2019 [12]	1.6	Yes	No	PO	SGE	Asymptomatic after two years
Torres, A., et al., 2021 [35]	1.4	No	No	PO	SGE	Bone healing evident after 12 months
Sônia, T.d.O., et al., 2018 [36]	2.7, 2.8	No	No	PO	SGE	Complete healing after one year
Fonseca, F.O., et al., 2020 [49]	1.1	No	Yes	PO	SGE	No symptoms after 12 months.
Tavares, W.L.F., et al., 2020 [50]	(a) 4.7 (b) 4.6 (c) 1.6	No Yes	Yes	PO	SGE	No symptoms after 12 months.
Maia, L.M., et al., 2020 [51]	4.6	Yes	No	PO	SGE	No symptoms after 12 months.
Freire, B.B., et al., 2021 [52]	1.1	No	Yes	PO	SGE	Complete healing after 24 months.
Doranala, S., et al., 2020 [53]	1.1	No	Yes	PO	DGE	Complete recovery with no symptoms after 12 months
Casadei, B.d.A., et al., 2020 [54]	1.5	Yes	No	PO	SGE	Complete recovery with no symptoms after 3 months
Maia, L.M., et al., 2022 [55]	2.1	Yes	Yes	RT	SGE	Healing in 18 months
Perez, C. 2020 [56]	1.6	Yes	No	RP	SGE	Periapical healing region at 1 year.
Strbac, G.D., et al., 2017 [42]	1.5	Yes	No	OP	SGE	Healing of periapical tissues after 12 months
Giacomino, C.M. 2018 [57]	1.7	Yes	No	AP	SGE	Asymptomatic after 12 weeks
Popowicz, W. 2019 [58]	2.5	Yes	No	OP	SGE	Asymptomatic after 7 months
Benjamin, G., et al., 2021 [59]	2.2	Yes	No	OT	SGE	Asymptomatic after 10 days.
Meda, R.G., et al., 2022 [43]	2.3	No	No	AUT	SGE	Bone completely healed after 24 months
Gambarini, G., et al., 2019 [60]	1.2	Yes	No	AP	SGE	Healing visible at three months with complete healing at 6 months
Villa-Machado, P.A., et al., 2022 [61]	3.1	Yes	No	OP	SGE	Asymptomatic after 8 weeks
Connert, T., et al., 2018 [62]	4.1	Yes	No	OP	SGE	Asymptomatic after 12 weeks
Torres, A., et al., 2019 [63]	2.3	No	Yes	OP	SGE	Periapical healing at 6 months
Silva, A.S., et al., 2020 [64]	2.2	No	Yes	OP	SGE	Complete healing after one year
Santiago, M.C., et al., 2022 [65]	4.5	No	No	OP	SGE	Asymptomatic after 6 months
Krug, R., et al., 2020 [18]	3.6	No	Yes	OP	SGE	An average 3.6-fold reduction in lesion size as measured by optical coherence tomography after one year
Kaur, G. 2021 [66]	2.4	Yes	No	OP	SGE	Asymptomatic after 2 weeks.
Ali, A., et al., 2022 [26]	2.2	Yes	Yes	OP	SGE	Complete healing after a year.
Pujol, M.L., et al., 2021 [67]	4.4	No	Yes	OP	SGE	Asymptomatic after one year
Yan, Y.Q., et al., 2021 [68]	2.1	No	No	OP	SGE	Asymptomatic after two years
Mena-Álvarez, J., et al., 2017 [69]	2.7	Yes	No	OP	Dens evaginatus	Asymptomatic after one year

## 4. Discussion

Once the relevant papers have been identified and described, various guided endodontics applications can be considered.

**Openings used in endodontics:** The initial stage in non-surgical root canal treatment is an endodontic access cavity, and much of the published research is based on this procedure. Four studies examined both minimally invasive and ultraconservative treatments [38,45,70,71,72]. Tooth structure can be conserved and instrument stress can be minimized by using a linear, ultra-conservative approach as described by Gambarini G et al. [38,73]. Access canals in endodontics are a contentious subject due to the variety of ways in which they can be defined and categorized [74,75,76]. While the traditional approach calls for removing the pulp chamber lid and going straight to the coronal third of the canals, the conservative and ultra-conservative approaches require entering through the central fossa and barely widening it to see where the canals are (in which the minimal access is made in the deepest center of the tooth). By keeping the incisions straight and the movements unimpeded when working in the coronal third, one can reduce the likelihood of perforations, false passages, and transferred canals [74,75]. This will help the operators in conserving the tooth structure. By using a conservative approach, the amount of healthy tooth structure that needs to be removed is minimized, which helps to preserve the overall strength and integrity of the tooth. Because of the trajectory produced by minimally invasive cavities, endodontic instruments must often flex in order to apply tension to the canal. Accidents caused by medical personnel, such as fractures and falls, are possible outcomes [14]. However, DGE prevents this by providing a direct, straight, and parallel path to the canal axis. In the author’s opinion, this will provide conservative access to the root canal orifices and endodontic treatment can be performed conservatively. Despite being minimally invasive, conservative access cavities provide adequate access to the pulp chamber and root canals, which facilitates the efficient removal of infected or damaged tissue and the thorough cleaning and shaping of the canals. It has been shown that less tooth tissue is lost as compared to “standard access cavities”. Even while research has shown that the fissure resistance of posterior teeth might vary based on the endodontic access method utilized, the resistance of anterior teeth is consistently high regardless of the approach taken. For teeth with intact marginal ridges, endodontic access has been shown in investigations to have no influence on tooth stiffness. Data collected by a dynamic guided navigation system were used by Simon GC et al. [77]. In this study, instead of using drills to create multiple access cavities, CO_2_ laser ablation was employed in both open surgery and minimally invasive surgery. It is possible to reduce contamination in the outer layers of soft or hard tissues by burning the contaminated tissues and toxins to a high degree using lasers. Hemostasis induction is also a useful technique for dealing with pulmonary resections [45]. However, it is important to remember that the heat produced by the laser heat might produce pulp hyperemia and that this can be avoided by eliminating the afflicted pulp tissue using hand equipment [44]. Laser-assisted pulpotomies had similar clinical and radiographic success rates as those of mineral trioxide aggregate and formocresol. Since the digital representation of the occlusal surface is provided by the dynamic navigation system, CBCT data are not required for laser surgery. The user can program the computer-controlled laser to automatically complete a predetermined access procedure. Due to its potential use in difficult situations with calcifications, guided endodontics may be useful for patients with atypical tooth morphologies that make conventional endodontic treatments difficult. While 3D splints have been mentioned in the treatment of dens invaginatus and dens evaginatus by several researchers, no studies have focused on this technique. Utilizing a non-invasive cavity preparation method, Jain SD et al. drilled at high speeds and used DGE to identify calcified canals. In cases when these canals were manually located, there was a higher incidence of perforations and tissue loss [41]. During SGE, the treatment time was reduced by switching from low-velocity to high-velocity drills and using 3D guides. This study added more weight to the argument that DGE use is an acquired skill due to the constant demands it places on one’s hand-eye coordination and spatial awareness. This confirms what was found in a previous study by Connert T et al., the efficacy of DGE therapy ultimately rests on the expertise of an operator who is immune to the effects of DGE. The authors believe that the precision provided by the DGE will allow the operator to accurately plan the treatment and guide the instruments to the exact location of the canal. This can lead to improved accuracy and efficiency of the procedure, resulting in better outcomes.

**Detecting calcified canals**: Extracting a small quantity of tooth tissue, and finishing the course of treatment in about the same length of time are all things that a non-specialist operator may be able to achieve just as well as an endodontic specialist, as shown in the study by Connert T et al. Because the access cavities created by GE, especially SGE, can only provide linear access, this treatment modality cannot be utilized on teeth that include curved canals or unusual shapes. Multiple guides for direct gonadectomized endodontics will need to be fabricated if a single tooth has more than one straight canal. In cases where many neighboring teeth, such as incisors, require root canal therapy, options such as a single guide with multiple accesses or a DGE treatment scheduled for a single appointment to complete all of the root canals can be considered. Teeth with PO and relatively straight or slightly curved apical third canals are suitable for root canal therapy. Fonseca WL used a combination of SGE/DGE to the extent possible, the traditional instrumentation of the curved region, and retrenching as photodynamic therapy. Therefore, it is recommended that further research should be conducted on a variety of problems. However, the overall understanding is that the use of guided endodontics allows for the operator to precisely control the instruments and navigate around the curves of the canal with minimal risk of damaging the surrounding tissues. It can also reduce the risk of complications such as instrument fracture, perforation, or ledge formation in curved canals, as it allows for a more controlled and minimally invasive approach to treatment.

**Apicoectomy and osteotomy** are two particularly notable surgical options. The second most common GE operation after approximation is osteotomy [42]. Only one retrospective study provides long-term follow-up data on patients who underwent SGE [78]. However, the SGE procedure is not disclosed. More studies of patients who undergo either sort of GE are needed, but with longer follow-up durations. All the studies produced flaps analogous to those employed in clinical settings of endodontic microsurgery. Several studies have suggested avoiding the flap procedure when removing the masticatory palatal mucosa [64]. This novel approach to endodontic microsurgery may lessen the risk of complications and post-operative discomfort when apicoectomies on the palatal roots of maxillary molars are performed [66].

**Glass fiber post removal:** Taking apart the glass fiber post, extraction of a post and crown from a previously treated tooth is the third most common procedure while utilizing GE [56]. This is studied by Afram A et al. When other methods of removal have failed, ultrasonic tips are frequently used to remove fiberglass posts [67]. One risk is a punctured tooth [79]. Additionally, due to the similarity in the color of the post and the dentine, this procedure can cause some problems. After the post is removed, the radicular canal can be widened depending on the dentist’s skill [41]. This is accomplished by removing extra dentine around the post. Because of these concerns, GE has been presented as a potential treatment option. Perez C et al. observed that the apical gutta-percha was reachable in 87.5 percent of treated teeth with SGE, with the remaining 12 percent being inaccessible due to root curvature. Notable as well is the fact that this study models the artefacts inherent to CBCT pictures, which can make it more challenging to recover design and construction data. However, it fulfilled the requirement, and in a quicker way than either the UTC recommendations or the lonthingem drills. Laser post removal is just as effective as milling and microscopy but much faster. However, familiarity with the new set-up was necessary before the operator could get to work swiftly and easily. The use of guided endodontics can help accurately position the drill or bur for removing the post. This is particularly useful when dealing with a broken post or a post that is deeply embedded in the root canal, as it allows for precise removal without causing additional damage to the tooth. It can also help reduce the risk of damage to the tooth or surrounding tissues during post removal. However, it is important to note that the success of post removal ultimately depends on the skill and experience of the dentist performing the procedure. While guided endodontics can enhance the efficacy of post removal, it is not a substitute for proper training and experience in performing endodontic procedures.

Guided endodontics is still in its infancy. Hence, the sorts of studies that can be found in the literature are limited [80]. This limits the scope of the current work. More studies, such as randomized clinical trials, are needed to assess the results of SGE compared to DGE in actual patients, with a focus on clinical and radiological outcomes throughout time. Although the selected publications were carefully reviewed, they contain bias due to the use of 3D-printed teeth, which lack the properties of natural teeth. Additionally, the follow-up periods in several papers were too short; in other cases, they lasted for only a few days or weeks, which is insufficient for establishing effective tactics.

## 5. Conclusions

The guided endodontics can be put to good use in the removal of fiberglass reinforcements, the placement of retrograde fillings, and the correction of morphologically abnormal teeth. Static-guided endodontics has many benefits, including its simplicity (almost anybody can perform it), its speed (less time in the dentist’s chair), and its precision (reduced risk of infection) in comparison to traditional endodontics. In the oral cavity, the lack of remarkable stability of partially edentulous patients, the time required to develop and manufacture 3D guides, and the fact that linear access only works for straight ducts are some of the drawbacks of static-guided endodontics. Digital-guided endodontics has several benefits over traditional endodontics, including being more ergonomic, the ability to make adjustments and reposition instruments in real-time, greater accuracy due to fewer design errors, and the possibility of being used in cases of multi-rooted teeth.

## Figures and Tables

**Figure 1 medicina-59-00678-f001:**
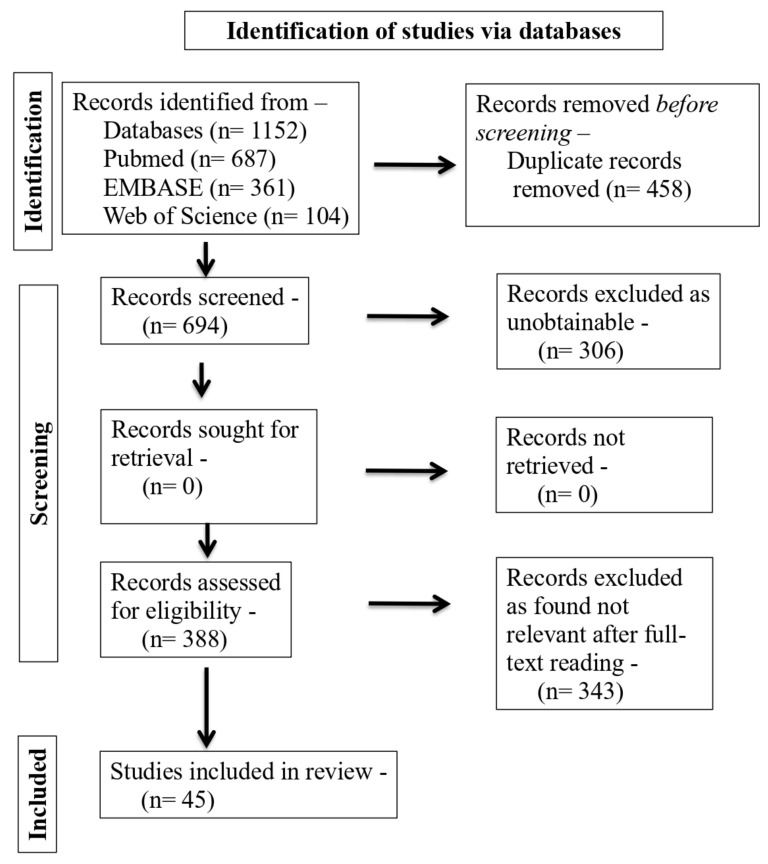
PRISMA flowchart of the study selection process.

## Data Availability

Not applicable.

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
