# Peer review of "Guided Endodontic Surgery: A Narrative Review"

_medicina, 2023, doi:10.3390/medicina59040678_

Round 1

Reviewer 1 Report (Previous Reviewer 2)

Dear authors. I have no further comments.

Author Response

Comment: Dear authors. I have no further comments.

Response:  Thank you and noted.

Reviewer 2 Report (New Reviewer)

It is well designed and written article.

Author Response

Comment:  It is well designed and written article.

Response: Thank you and noted.

Reviewer 3 Report (New Reviewer)

Guided endodontics is a relatively new trend in endodontics. Guided endodontic surgery is only one part of this topic. According to the article's title, the authors must concentrate on this topic exclusively. Title of the paper and background do not correspond to the methodology, results and discussion part. The terms used as keywords for searching articles overwhelm a greater understanding of the theme, but this needs to be reflected in the article's title and abstract.

In the discussion part, I would like to see more authors' opinions on their chosen topic, not only the delivery of facts from the articles.

Author Response

Dear Editor and Reviewers,

We would like to thank you for the valuable comments and suggestions on the report “Guided Endodontic Surgery: A Narrative Review".  We believed that all these had positively contributed towards improving the quality of the report itself and hope that these will help in getting the report to be published in the " Medicina" journal.

I am resubmitting our revised manuscript after addressing point-by-point, the comments and suggestions from the reviewers. We also would like to thank you in advance for agreeing to review the resubmitted version of the manuscript.

Thank you.

Dr. Farooq Ahmad Chaudhary

School of Dentistry, SZABMU.

Response to comments of Reviewer 1

Comment: Dear authors. I have no further comments.

Response:  Thank you and noted.

Response to comments of Reviewer 2

Comment:  It is well designed and written article.

Response: Thank you and noted.

Response to comments of Reviewer 3

Comment: According to the article's title, the authors must concentrate on this topic exclusively.

Response: Thank you for your valuable feedback. The topic of the paper is “Guided Endodontic Surgery”, and  objective of this paper is to review recent scientific literature related. The literature search and manuscript writeup has been done keeping the topic in mind. The manuscript is thoroughly checked again and revised where necessary as suggested by reviewer.

Comment: Title of the paper and background do not correspond to the methodology, results and discussion part.

Response: The authors have focused on the “ Guided Endodontic Surgery” aspect.

In the methodology, the search terms have been mentioned keeping in mind the topic of the study. Changes have been done as suggested. In the results section, it is mentioned that the unrelated papers were excluded from the study as the scope of the review was “Guided endodontic surgery”. The shortlisted studies were only 45 at the end which were related to the study.

Comment: The terms used as keywords for searching articles overwhelm a greater understanding of the theme, but this needs to be reflected in the article's title and abstract.

Response: Key words have been modified and revised, keeping in mind the title of the study as suggested by the reviewer.

Comment: Needs more authors' opinions on their chosen topic, not only the delivery of facts from the articles.

Response: The discussion has been revised and modified as recommended by the reviewer.

This manuscript is a resubmission of an earlier submission. The following is a list of the peer review reports and author responses from that submission.

Round 1

Reviewer 1 Report

1. as per the title is this article a literature review or a systematic review? please use the correct format.

2. Clarify whether a literature review is any study?

3. Mention what types of articles are included in the review. case reports/study/ both?

4. rewrite the introduction with more clarity. write each part separately ex. CBCT, static guides, dynamic guides etc . 

5. rewrite methodology if it is a systematic review. if the literature review methodology section should be eliminated.

6. follow Prizma guidelines if it is a systematic review

7. please mention whether all the articles included were studies/ case reports

9. please do English and grammar corrections

10. the content included in the discussion is not as per title of the article.

Reviewer 2 Report

The authors have conducted a review on the topic of guided endodontics surgery. However the limitations of the present manuscript are so strong that it makes difficult to make recommendations to upgrade it. I believe the authors need to completely reformulate the study design and definitely assume if this is a systematic review or a narrative review.  

I suggest the authors to place the keywords by alphabetic order. The end of the introductions lacks of a proper aim sentence.

I find myself lost on this study design. The authors show a kind of systematized search, but that lack of the most relevant points of the Systematic Review (please check the PRISMA statement for a better understanding), but them state in the title it is a Literature Review (aka Narrative Review). If this is a systematic review all the technical issues such as databases selected, manual search, previous studies references assessment, consultation of reference journals, gray literature assessment, contact with previous study authors, dates of search, data extraction methods, scientific merit assessment, assessment of bias, registration of the review, so on, son on… is just missing. If this is a solo narrative review… why are the authors showing a methodology for the search? Additionally, the authors present the Discussion on a Result presentation manner instead a true Discussion.

I suggest the authors to completely re-prepare the manuscript and submit it again, as a new manuscript, after clarifying all this issues.